# Molecular Characterization of a Clade 2.3.4.4b H5N1 High Pathogenicity Avian Influenza Virus from a 2022 Outbreak in Layer Chickens in the Philippines

**DOI:** 10.3390/pathogens13100844

**Published:** 2024-09-28

**Authors:** Zyne Baybay, Andrew Montecillo, Airish Pantua, Milagros Mananggit, Generoso Rene Romo, Esmeraldo San Pedro, Homer Pantua, Christina Lora Leyson

**Affiliations:** 1BioAssets Corporation, Santo Tomas 4234, Batangas, Philippines; zyne.baybay@bioassets.com.ph (Z.B.); andrew.montecillo@bioassets.com.ph (A.M.); airish.gatlabayan@bioassets.com.ph (A.P.); homer.pantua@bioassets.com.ph (H.P.); 2Institute of Biological Sciences, University of the Philippines Los Baños, Los Baños 4031, Laguna, Philippines; 3Department of Agriculture Regional Field Office III, San Fernando 2000, Pampanga, Philippines; milagros.mananggit@rfo3.da.go.ph; 4Philippine College of Poultry Practitioners, Tanay 1980, Rizal, Philippinesesmi.sanpedro@cobbvantres.com (E.S.P.); 5Balik Scientist Program, Department of Science and Technology, Bicutan 1631, Taguig, Philippines

**Keywords:** high-pathogenicity avian influenza, chickens, whole-virus-genome sequencing, Goose/Guangdong lineage, clade 2.3.4.4b

## Abstract

H5 subtype high-pathogenicity avian influenza (HPAI) viruses continue to devastate the poultry industry and threaten food security and public health. The first outbreak of H5 HPAI in the Philippines was reported in 2017. Since then, H5 HPAI outbreaks have been reported in 2020, 2022, and 2023. Here, we report the first publicly available complete whole-genome sequence of an H5N1 high-pathogenicity avian influenza virus from a case in Central Luzon. Samples were collected from a flock of layer chickens exhibiting signs of lethargy, droopy wings, and ecchymotic hemorrhages in trachea with excessive mucus exudates. A high mortality rate of 96–100% was observed within the week. Days prior to the high mortality event, migratory birds were observed around the chicken farm. Lungs, spleen, cloacal swabs, and oropharyngeal–tracheal swabs were taken from two chickens from this flock. These samples were positive in quantitative RT-PCR assays for influenza matrix and H5 hemagglutinin (HA) genes. To further characterize the virus, the same samples were subjected to whole-virus-genome amplification and sequencing using the Oxford Nanopore method with mean coverages of 19,190 and 2984, respectively. A phylogenetic analysis of the HA genes revealed that the H5N1 HPAI virus from Central Luzon belongs to the Goose/Guangdong lineage clade 2.3.4.4b viruses. Other segments also have high sequence identity and the same genetic lineages as other clade 2.3.4.4b viruses from Asia. Collectively, these data indicate that wild migratory birds are the likely source of H5N1 viruses from the 2022 outbreaks in the Philippines. Thus, biosecurity practices and surveillance for HPAI viruses in both domestic and wild birds should be increased to prevent and mitigate HPAI outbreaks.

## 1. Introduction

High-pathogenicity avian influenza virus (HPAIV) is a type-A influenza virus that causes severe disease in gallinaceous birds such as chickens, turkeys, and quail. Like other avian influenza viruses (AIVs), HPAIV has a broad host range and can infect a wide range of bird and mammalian species [1]. HPAIV outbreaks have incurred huge economic losses. Additionally, HPAIV outbreaks have also raised public health concerns due to the virus’ ability to infect and cause fatality among humans [2]. The control of HPAIV outbreaks is performed through biosecurity, surveillance, systemic culling of infected flocks, and, in some cases, vaccination.

H5Nx HPAIVs from clade 2.3.4.4b have circulated in wild bird populations in the past several years after becoming prevalent in Europe and Asia in 2016 [3]. The clade 2.3.4.4b viruses have caused worldwide outbreaks since 2021 in Europe, Africa, Asia, and the Americas [4]. Between 2022 and 2023, Philippines, Japan, South Korea, Taiwan, Hongkong, Vietnam, Indonesia, China, and other Asian countries have confirmed H5N1 outbreaks in both domestic birds and wild birds [5].

In the Philippines, the Department of Agriculture reported the first HPAIV outbreak in July 2017 when it was detected from commercial poultry farms in the Central Luzon Region III. More H5Nx subtype HPAIV outbreaks occurred in 2020, 2022, and 2023 [6]. Despite multiple HPAIV outbreaks over the past years, genetic information on viruses that caused outbreaks in the Philippines is lacking. It is therefore difficult to ascertain the origins of these viruses. To our knowledge of publicly available information, here we report the first complete whole-genome sequence and lineage analysis of an H5N1 subtype HPAIV isolate from chickens in the Philippines.

## 2. Materials and Methods

### 2.1. Sample Preparation and Quantitative RT-PCR

Lungs, spleen, cloacal swabs, and oropharyngeal–tracheal swabs were collected from two birds namely, patay ‘PTY’ and mahina ‘MHN’, which were found dead or weak, respectively. For each bird, nucleic acid extraction was performed on the spleen, cloacal swabs, pooled lungs and oropharyngeal–tracheal swab samples using a Patho Genespin DNA/RNA Extraction Kit (iNtRON Biotechnology, Gyeonggi-do, Republic of Korea) following the manufacturer’s instructions, with a heat inactivation step prior to lysis. RNA samples were quantitative-RT-PCR-positive for influenza A matrix RNA and H5 RNA using the virotype Influenza A kit (Indical, Leipzig, Germany) and the virotype Influenza A H5/H7/H9 RT-qPCR kit (Indical, Leipzig, Germany), respectively. The eight influenza A segments were then amplified using published integrated molecular indexing primers [7] and SuperScript™ IV One-Step RT-PCR System kit (Thermo Fisher Scientific, Waltham, MA, USA). PCR products were purified with QIAquick PCR Purification kit (Qiagen, Hilden, Germany) and checked for quality and concentration using DeNovix DS-11 Spectrophotometer (DeNovix, Wilmington, DE, USA).

### 2.2. Whole-Genome Sequencing

For DNA library preparation, about 1 μg of purified PCR amplicon from each sample was used as input to the Ligation sequencing kit 1D SQK-LSK109 (Oxford Nanopore Technologies, England, UK) following the manufacturer’s protocol for ligation sequencing amplicons. About 50 fmol of the resulting DNA library was loaded onto a MinION flowcell (MIN106 R9.4.1; ONT, England, UK) and was run in MinKNOW v. 23.07.15 for up to 24 h. Reads were base-called using a super accurate model (dna_r9.4.1_450bps_sup.cfg), deconvoluted in Porechop v0.2.4 (https://github.com/rrwick/Porechop, accessed on 2 May 2024) with default parameters (-discard_middle) and loaded on Geneious Prime (v.2023.2.1; Biomatters Ltd., Auckland, New Zealand).

### 2.3. Sequence Assembly

Reads were mapped initially using the default Geneious mapper (medium sensitivity/fast, 5 iterations) against a subset of avian influenza virus genomes in Asia in the past three years (Type: A, Region: Asia, Segment: Any, Subtype: H5 and N (any), Released from 2020 to 2023) obtained from the National Center for Biotechnology Information (NCBI) Influenza Virus Resource (https://www.ncbi.nlm.nih.gov/genomes/FLU/, accessed on 2 May 2024). The consensus sequence was taken from this first round of reference-based mapping (65% consensus threshold) and submitted to BLASTn (https://blast.ncbi.nlm.nih.gov/Blast.cgi, accessed on 2 May 2024). The top hit was used as the reference sequence for the second round of read mapping. Manual inspection of the final consensus sequences was conducted to resolve any single-base ambiguities. The nucleotide sequences of PTY and MHN samples were deposited to the NCBI GenBank database with accession numbers PP732683–PP732698 and to the Global Initiative on Sharing All Influenza Data (GISAID) database with accession numbers EPI_ISL_19057794 (PTY) and EPI_ISL_19061458 (MHN).

### 2.4. Phylogenetic Analysis

For phylogenetic analysis, all full-length sequences of H5 clade 2.3.4.4/b/c/e/g/h from avian hosts and from any location were downloaded from Global Initiative on Sharing All Influenza Data (GISAID) Database [8]. All sequences from Asia and from avian hosts of any subtype were downloaded for segments other than HA. For each segment, sequences with duplicate names were removed, and the dataset was down-sampled at the following thresholds using the CD-HIT version 4.8.1 [9]: 96% for PB2, PB1, PA, and NA; 99% or 98% for H5; 97% for NP and NS; and 97.5% for M. The following number of sequences were thus obtained out of the total number of downloaded sequences for the PB2, PB1, PA, HA, NP, NA, M, and NS segments, respectively: 410/14967, 313/11408, 174/5185, 443/12423, 481/14920, 565/16973, 407/15686, and 416/15726. Added to this dataset are the sequences from the top BLAST hits, namely A/crow/Miyagi/TU69-55/2023 (GenBank accession no. LC765306-LC765313), A/crow/Fukuoka/TU48-37/2022 (GenBank accession no. LC765290-LC765297), and A/feline/South Korea/SNU1/2023 (GenBank accession no. OR680853, OR680855, OR680857, OR388761, OR388763-OR388766); G2 group sequences as reported by Takadate et al. [10], namely A/duck/Bangladesh/19D1874/2022, A/white-tailed eagle/Hokkaido/20220210001/2022, A/white-fronted goose/Miyagi/0410D001/2022, A/Eurasian_wigeon/Hokkaido/Q71/2022, A/peregrine falcon/Kanagawa/1409C001T1/2022, A/slaty-backed_gull/Hokkaido/0111M111/2022, A/peregrine falcon/Niigata/1510C001T/2022, A/large-billed crow/Niigata/1503B017/2023, A/chicken/Kagoshima/22H4T/2022, A/chicken/Nagasaki/22A6T/2022, A/chicken/Saitama/22A10T/2022, A/chicken/Chiba/21B5T/2022; reference sequences as defined by Fusaro et al. [11]; and the sequences obtained from the local HPAIV samples in this study. The sequences were then aligned using MAFFT version 7.407 [12], and maximum likelihood trees were created using RaxML-ng version 1.2.1 with the GTR+FO+G4m model and automatic bootstrapping [13]. A complete list of reference sequences and their associated list accession numbers or isolate IDs can be found in Appendix A. Phylogenetic trees were visualized and annotated using TreeViewer v.2.2.0 [8].

## 3. Results

An HPAIV outbreak occurred in layer chickens from a farm in Central Luzon (Region III), Philippines. The chickens exhibited signs of lethargy, droopy wings, and ecchymotic hemorrhages in trachea with excessive mucus exudates. As is typical for HPAIV in chickens, high mortality was observed at 96–100%. It was also noted that prior to the outbreak, wild birds such as egrets were seen around the poultry housing (personal communication with farm personnel).

Two samples from the chickens were sequenced and analyzed. Prior to whole-genome sequencing, the pooled lungs and oropharyngeal–tracheal swabs from two chicken samples referred to as patay (PTY) and mahina (MHN) were initially screened for the presence of Influenza A RNA with C_q_ values of 18 and 19 for pooled lungs and oropharyngeal–tracheal swabs, 21 and 24 for cloacal swabs, and 19 and 21 for spleen samples, respectively. The PTY and MHN samples were obtained from a dead and sick bird, respectively. A subsequent subtyping multiplex assay on the pooled lungs and oropharyngeal–tracheal swabs confirmed the presence of Influenza A subtype H5 from both the PTY (C_q_ = 19) and MHN (C_q_ = 21) samples. All eight gene segments were successfully amplified through RT-PCR using universal influenza primers and processed for high-throughput sequencing.

After quality control and trimming, a total of 515,834 and 101,263 reads were obtained and used on the map-based assembly and consensus sequence determination from the PTY and MHN samples designated as A/chicken/Philippines/BA-PTY/2022|H5N1 (PTY) and A/chicken/Philippines/BA-MHN/2022|H5N1 (MHN), respectively (Appendix A). Each of the eight segments were sequenced at >500× mean coverage except for PB1 and PA genes of MHN.

BLAST results (Table 1) indicate that the sequence of seven out of the eight segments had >99% nucleotide identity similarity with the H5N1 sequences isolated from crows in Japan between 2022 and 2023, while the NS sequence is most similar to an H5N1 strain detected from a domestic cat in South Korea at >99% nucleotide identity.

The phylogenetic analysis of the PTY and MHN HA sequences showed that these virus isolates belong to the Goose/Guangdong lineage (Gs/Gd) clade 2.3.4.4b (Figure 1). Specifically, the PTY and MHN HA genes are most related to the 2022–2023 isolates from Asia, Europe, North America, and South America. In all the segments, the PTY and MHN cluster with top BLAST hits, namely A/crow/Miyagi/TU69-55/2023 (H5N1), A/crow/Fukuoka/TU48-37/2022(H5N1), A/feline/South Korea/SNU1/2023(H5N1) (Appendix A). We additionally observed that PTY and MHN cluster with the G2c group [10,14,15] of clade 2.3.4.4b viruses (Figure 2D). The G2c group of clade 2.3.4.4b viruses were isolated in Japan during the 2022–2023 season [10]. PTY and MHN have a genotype most similar to G2c group viruses: A/chicken/Nagasaki/22A6T/2022 and A/large-billed crow/Niigata/1503B017/2023. Relative to other G2c group viruses, reassortments in PB2, PB1, PA, NP, and NS were observed in PTY and MHN (Figure 2, Appendix A).

We also compared PTY and MHN to the genotypes of 2020–2022 clade 2.3.4.4 viruses found in Europe using the reference sequences as identified by Fusaro et al. [11]. These reference sequences encompassed different AIV subtypes and various geographical locations that included Europe, Asia, and Africa. We found that PTY and MHN share a common ancestor with only the following reference sequences and their corresponding segments: A/environment/Bangladesh/17E82/2021 (H6N1) at the NP segment (Figure 2E) and A/turkey/England/057679/2021 (H5N1) at the NA and M segments (Figure 2F,G). Since this a genotype constellation is not described in [11], the genotype of PTY and MHN therefore appears to be distinct compared to 2020–2022 clade 2.3.4.4 viruses found in Europe. This observation highlights the broad diversity of clade 2.3.4.4 viruses and shows that multiple genotypes of clade 2.3.4.4 viruses simultaneously circulate across the globe.

Both PTY and MHN had the PLREKRRKR↓GLF multi-basic motif at the HA cleavage site. This motif has been reported in many HPAI viruses found in poultry, wild birds, and in humans, as well as in the Gs/Gd lineage of H5 HPAI viruses [16]. Additionally, we found several molecular markers in PTY and MHN that are associated with virulence in avian or mammalian species (Appendix A). Notably, we did not find amino acid sequence changes E627K/V or D701N/V, which are associated with adaptation to mammalian species [17,18,19,20,21]. We also found M105V, which is associated with adaptation of duck-derived AIVs to chickens [22].

## 4. Discussion

Taking the sequence analyses and observation of wild birds around the premises, it is therefore possible that the H5N1 was brought into the Philippines by migratory birds. Other pathways for the introduction of the H5N1 HPAIVs, such as transport of infected birds, are also possible and warrant further investigation.

Central Luzon or Region III, where PTY and MHN were sampled from, is known for its high concentration of domestic chickens and ducks. At the same time, Central Luzon is also an important location for wild migratory birds. Specifically, the Candaba swamp in Central Luzon is an ecologically important area that hosts wild migratory birds and, along with another municipality, houses most of the poultry farms in the province of Pampanga, Central Luzon [23]. It is thus believed that the overlap of domestic and wild bird populations in the Candaba swamp plays an important role in the importation and spread of HPAIV in the Philippines. Indeed, the province of Pampanga has been shown to carry the highest risk of HPAIV outbreaks, specifically in poultry farms in the municipalities of San Luis and Candaba [23].

This study is not the first case reported in Central Luzon, which has had a history of H5 HPAIV outbreaks in 2017, 2020, and 2022–2023 [6]. Subsequent report of H5 HPAIV sequences from 2019 to 2020 were also from Central Luzon and in the province of Pampanga. These virus isolates belong to Gs/Gd clades 2.1.3.2 (H5N8), 2.3.4.4e (H5N6), and 2.2.1 (H5Nx) (Appendix A). The observation that these viruses are distinct from the 2022 H5N1 clade 2.3.4.4b viruses reported herein demonstrates that several incursions have occurred over the past several years in the Central Luzon region. This further highlights the importance of Central Luzon in the epidemiology of H5 HPAIVs in the Philippines. Most importantly, there is thus a need to increase biosecurity in domestic bird premises and heighten disease surveillance for both domestic and wild birds in the Central Luzon region. Such surveillance activities can shed light into the genetic origins of avian influenza viruses circulating in the Philippines and would inform the most appropriate intervention strategies using a molecular epidemiological approach. This study also highlights that H5 HPAI viruses are a global concern and the interconnectedness of HPAI virus disease ecology across vast geographical regions. 

## Figures and Tables

**Figure 1 pathogens-13-00844-f001:**
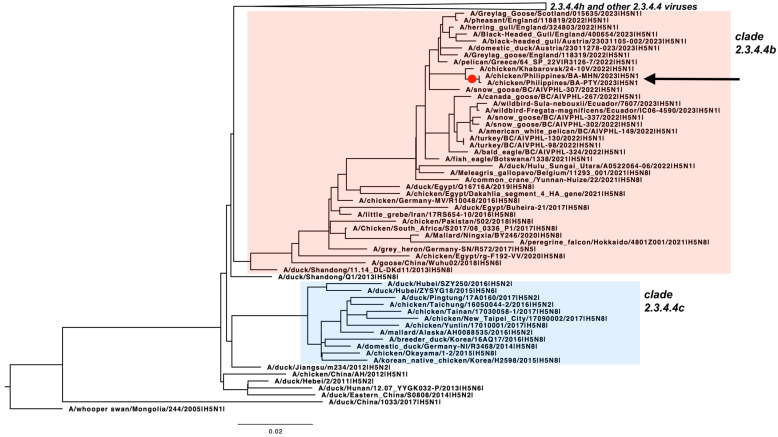
H5 phylogeny reveals that the HA genes from a virus detected in Central Luzon, Philippines belong to the Gs/Gd clade 2.3.4.4b. A/chicken/Philippines/BA-PTY/2022|H5N1 and A/chicken/Philippines/BA-MHN/2022|H5N1 are indicated by a black arrow and red dot. All hemagglutinin (HA) sequences from the H5 clade 2.3.4.4/b/c/e/g/h from avian hosts and from any location were downloaded from GISAID Database. After removal of sequences with duplicated strain names, the dataset was down-sampled at 98% threshold using CD-HIT version 4.8.1. to obtain another dataset of 104 sequences. This dataset and the HA sequences obtained from the HPAIV samples from Central Luzon, Philippines were aligned using MAFFT v7.407 and maximum likelihood trees were created using RaxML-ng v1.2.1.

**Figure 2 pathogens-13-00844-f002:**
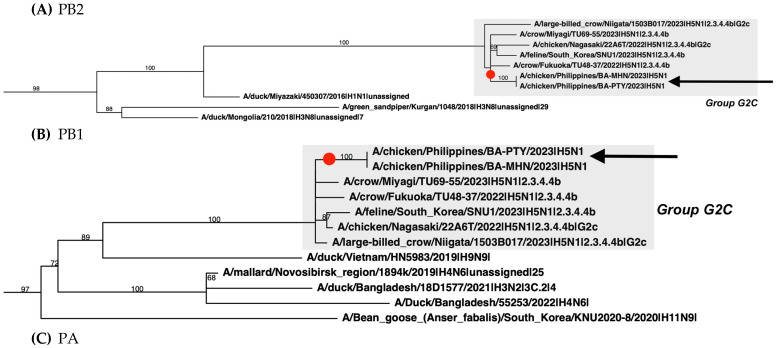
Viruses detected from Central Luzon, Philippines belong to group G2c viruses across all segments. A/chicken/Philippines/BA-PTY/2022|H5N1 (PTY) and A/chicken/Philippines/BA-MHN/2022|H5N1 (MHN) are indicated by a black arrow and red dot. Avian influenza virus sequences were downloaded from GISAID and subsequently down-sampled with CD-HIT. Phylogenetic trees were generated using RaxML-ng v1.2.1 with alignments made using MAFFT v7.407. For simplicity, only branches showing PTY and MHN are shown. The complete phylogenetic trees are shown in Appendix A. (**A**) PB2, (**B**) PB1, (**C**) PA, (**D**) HA, (**E**) NP, (**F**) NA, (**G**) M, (**H**) NS.

**Table 1 pathogens-13-00844-t001:** Top BLAST hits for H5N1 strains of A/chicken/Philippines/BA-PTY/2022|H5N1 based on the GenBank Nucleotide database.

Segment	Size (bp)	GC Content (%)	Strain Name for Top BLAST Hit	Identity at Nucleotide Level (%)	GenBank Accession No.
PB2	2323	43.7	A/crow/Miyagi/TU69-55/2023(H5N1)	99.8	LC765306
PB1	2323	43.7	A/crow/Miyagi/TU69-55/2023(H5N1)	99.4	LC765307
PA	2214	44.9	A/crow/Miyagi/TU69-55/2023(H5N1)	99.6	LC765308
HA	1757	41.9	A/crow/Miyagi/TU69-55/2023(H5N1)	99.5	LC765309
NP	1544	47.3	A/crow/Fukuoka/TU48-37/2022(H5N1)	99.6	LC765294
NA	1436	44.1	A/crow/Fukuoka/TU48-37/2022(H5N1)	99.4	LC765295
M	1002	49.5	A/crow/Fukuoka/TU48-37/2022(H5N1)	99.9	LC765296
NS	864	43.1	A/feline/South Korea/SNU1/2023(H5N1)	99.3	OR388766

## Data Availability

The genome sequences of PTY and MHN samples generated from this study are openly available in the NCBI database with accession numbers PP732683–PP732698 and in the Global Initiative on Sharing All Influenza Data (GISAID) database with accession numbers EPI_ISL_19057794 (PTY) and EPI_ISL_19061458 (MHN).

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
