# Peer review of "Molecular Characterization of a Clade 2.3.4.4b H5N1 High Pathogenicity Avian Influenza Virus from a 2022 Outbreak in Layer Chickens in the Philippines"

_pathogens, 2024, doi:10.3390/pathogens13100844_

Round 1

Reviewer 1 Report

Comments and Suggestions for Authors

Dear authors

I hope all of you are always fine. Regarding the revision of the manuscript No. pathogens-3031520, titled “Molecular characterization of a clade 2.3.4.4b H5N1 high pathogenicity avian influenza virus from a 2022 outbreak in layer chickens in the Philippines”. Really it is an excellent and interesting research, however, many major comments should be replied.

Major Comments

1-    In the abstract:

-       Line 20, the mortality rate should be corrected to 96-100% as mentioned in line 120.

-       Line 21, please add spleen and cloacal swabs to sampling.

2-    In Materials and Methods:

-       Why didn’t you examine samples from spleen and cloacal swabs??? So, you can remove them and depend only on lung and oropharyngeal-tracheal swabs.

-       Please add subtitles to each paragraph in Materials and Methods for easier understanding to all readers of your paper.

3-    In results: 

-       A full explain to the postmortem (PM) findings of examined birds should be written.

-       You can add some PM pictures if you have.

-       Phylogenetic trees for the other 7 segments (PB2, PB1, PA, NP, NA, M, and NS) should be added.

4-    Your discussion needs more explanation: I recommend to separate results from discussion and to add more details about the clade 2.3.4.4b H5Nx in your country with regards to other Asian, African, European and American countries.

Author Response

Dear authors

I hope all of you are always fine. Regarding the revision of the manuscript No. pathogens-3031520, titled “Molecular characterization of a clade 2.3.4.4b H5N1 high pathogenicity avian influenza virus from a 2022 outbreak in layer chickens in the Philippines”. Really it is an excellent and interesting research, however, many major comments should be replied.

Major Comments

1-    In the abstract:

-       Line 20, the mortality rate should be corrected to 96-100% as mentioned in line 120.

-       Line 21, please add spleen and cloacal swabs to sampling.

Response: The mortality rate in line 20 was modified to 96-100% as suggested. Spleen and cloacal swabs were added to sampling (line 21-23). The statement was modified from: “Pooled lung samples and oropharyngeal-tracheal swabs were taken from two chickens from this flock” to “Lungs, spleen, cloacal swabs, and oropharyngeal-tracheal swabs were taken from two chickens from this flock.”

2-    In Materials and Methods:

-       Why didn’t you examine samples from spleen and cloacal swabs??? So, you can remove them and depend only on lung and oropharyngeal-tracheal swabs.

Response: During necropsy, spleen and cloacal swabs were taken in addition to lungs and oropharyngeal-tracheal swabs. We initially decided to process only oropharyngeal-tracheal swabs and lungs as a starting point. To address the reviewer’s comment, we have returned to our long-term storage and performed quantitative RT-PCR (qRT-PCR) data for matrix genes on the spleen and cloacal swabs. Spleen and cloacal swabs were also added in the Materials and methods, specifically in lines 66-67. Lastly, Cq values of spleen and cloacal swabs were provided in line 144-145 in the Results and Discussion.

-       Please add subtitles to each paragraph in Materials and Methods for easier understanding to all readers of your paper.

Response: Thank you for your suggestions. We have included the following subtitles: 2.1. Sample preparation and quantitative RT-PCR, 2.2 Whole genome sequencing, 2.3 Sequence Assembly, 2.4 Phylogenetic analysis

3-    In results: 

-       A full explain to the postmortem (PM) findings of examined birds should be written.

Response:  Thank for this comment. We agree with the reviewer that PM findings would be very informative. Unfortunately, a full postmortem was not conducted when the birds were submitted to the laboratory. Thus, the PM findings are not available.

-       You can add some PM pictures if you have.

Response:  Regrettably, there were no pictures taken during sample collection. Moving forward, we aim to have a thorough PM and pictures during sample collection as much as possible.

-       Phylogenetic trees for the other 7 segments (PB2, PB1, PA, NP, NA, M, and NS) should be added.

Response:  Phylogenetic trees of the other 7 segments are available as supplementary data (Supplementary Tables S3-10). We would like to keep the manuscript short since it is submitted as a short communication. Thus, the large phylogenetic trees for other segments have been placed in supplementary data.

4-    Your discussion needs more explanation: I recommend to separate results from discussion and to add more details about the clade 2.3.4.4b H5Nx in your country with regards to other Asian, African, European and American countries.

Response: Thank you for this comment. The reviewer makes a good point. However, as mentioned above, our intention is to submit this manuscript as a short communication, which typically has the results and discussion sections combined. For this reason, we have retained the version where results and discussion are combined.

With regard to the comparison of our sequences to those already in the database, we have updated the phylogenetic analysis to further identify its closest relatives. We opted to retain our analysis to sequences in Asia because (1) Asia has the most diversity in H5 HPAIV sequences and (2) the observation that PTY and MHN were nearly identical to virus isolates in Japan and Korea. We have observed that PTY and MHN sequences belong to the G2c group with reassortments in PB2, PB1, PA, NP, and NS. Please see Figure 1B, Supplementary Figure S1 and lines 174-180.

Reviewer 2 Report

Comments and Suggestions for Authors

In this study, the authors conducted a whole genome amplification and sequencing of a clade 2.3.4.4.b H5N1 HPAI virus from an outbreak among chickens in the Philippines. The obtained data are the first available whole genome sequences of an H5N1 HPAI virus for the Philippines despite precedents of this virus infection outbreaks in this region. Consequently, the genomic characterisation of local strains is beneficial and provides information about virus origin, lineages of the virus’ gene segments and molecular backgrounds of virus pathogenicity.

Based on your research findings, I have some specific comments and suggestions that could potentially enhance the manuscript and improve the understanding of your results:

First, in line #124, the complete spelling of the sample names' abbreviations should be given in line #63, where they were mentioned the first time.

Second, Table 1 presents intermediate technical data for further full-genome analysis. Since that, it characterizes the quality of sequencing results and is less informative for discussion. Consequently, I suggest removing it from the main body of the paper and adding it to the supplement.

Third, your H5N1 viruses were detected in 2022; hence, the BLAST results in Table 2 indicate nucleotide identity predominantly with the crow isolates of 2023 from Japan and the feline strain from South Korea. These top BLUST hit strains were isolated a year after the H5N1 HPAIV outbreak in the Philippines and cannot be donors of the virus gene segments or serve as ancestors of this studied virus. Based on this, I suggest searching for BLAST hit strains among viruses detected in 2022 or previous years. If this suggestion is adopted, the sentences in lines #108, #143, and #156-157 must be revised.

There is a lack of information about the molecular characteristics of the viral gene segments, although the article's topic is “ Molecular characterization of a clade 2.3.4.4.b ….”. Molecular characteristics are limited to amino acid motifs at the HA cleavage site. Providing molecular markers of pathogenicity in other gene segments or unique mutations will be more informative.

In addition, information must be provided about wild bird species that were seen before the outbreak, how they are regularly seen in farmhouse areas, their mortality, etc.

Author Response

In this study, the authors conducted a whole genome amplification and sequencing of a clade 2.3.4.4.b H5N1 HPAI virus from an outbreak among chickens in the Philippines. The obtained data are the first available whole genome sequences of an H5N1 HPAI virus for the Philippines despite precedents of this virus infection outbreaks in this region. Consequently, the genomic characterisation of local strains is beneficial and provides information about virus origin, lineages of the virus’ gene segments and molecular backgrounds of virus pathogenicity.

Based on your research findings, I have some specific comments and suggestions that could potentially enhance the manuscript and improve the understanding of your results:

First, in line #124, the complete spelling of the sample names' abbreviations should be given in line #63, where they were mentioned the first time.

Response: Complete spellings of the abbreviated names, PTY and MHN, are provided in line 65 and 143, as suggested.

Second, Table 1 presents intermediate technical data for further full-genome analysis. Since that, it characterizes the quality of sequencing results and is less informative for discussion. Consequently, I suggest removing it from the main body of the paper and adding it to the supplement.

Response: Table 1 will be moved as a supplementary material, as suggested.

Third, your H5N1 viruses were detected in 2022; hence, the BLAST results in Table 2 indicate nucleotide identity predominantly with the crow isolates of 2023 from Japan and the feline strain from South Korea. These top BLUST hit strains were isolated a year after the H5N1 HPAIV outbreak in the Philippines and cannot be donors of the virus gene segments or serve as ancestors of this studied virus. Based on this, I suggest searching for BLAST hit strains among viruses detected in 2022 or previous years. If this suggestion is adopted, the sentences in lines #108, #143, and #156-157 must be revised.

Response: We thank the reviewer for this valid comment. We have performed additional phylogenetic analysis to show that PTY and MHN are G2c group viruses that were detected in 2022-2023 season in Japan (lines 174-180). Avian influenza seasons often straddle two years, similar to that observed in humans. Moreover, AIV prevalence often coincide with wild bird migration patterns. It is thus not uncommon to have a group of viruses isolated from two consecutive years as we observed for PTY, MHN, and their top BLAST hits.

Top BLAST hits, A/crow/Fukuoka/TU48-37/2022, A/crow/Miyagi/TU69-55/2023, and A/feline/South Korea/SNU1/2023 were isolated on 2022-Dec-26, 2023-Jan-28, and 2023-Jul respectively, while our PTY and MHN viruses were isolated in 2022-July-08. While these dates are 6-12 months apart, it is still within the range of a typical year in the avian influenza season. We acknowledge that this association is speculative but as of date, not much is known about the seasonality of avian influenza virus prevalence in the Philippines or the migration patterns of wild birds in the Philippines relative to the rest of the globe. This can be further teased apart with more surveillance and sequence activities, which the authors are currently continuing to pursue.

There is a lack of information about the molecular characteristics of the viral gene segments, although the article's topic is “ Molecular characterization of a clade 2.3.4.4.b ….”. Molecular characteristics are limited to amino acid motifs at the HA cleavage site. Providing molecular markers of pathogenicity in other gene segments or unique mutations will be more informative.

Response: Thank you for this comment. We have added an additional Supplementary Table (S3) and text on lines 195-200 that details the molecular markers found in the 2022 H5N1 isolates from the Philippines.

In addition, information must be provided about wild bird species that were seen before the outbreak, how they are regularly seen in farmhouse areas, their mortality, etc.

Response: Wild birds, such as egrets were seen around the farm by the caretakers (personal communication). This detail has been added to lines 138-139. We regrettably do not have additional information about the wild birds seen around the farm and the frequently with which these birds are seen. As mentioned above, the authors are hoping to continue projects along this line of investigation.

Reviewer 3 Report

Comments and Suggestions for Authors

The authors have analysed the whole genome sequence of HPAIV H5N1 2344b from two chickens of an outbreak in layers in central Luzon, Philippines. They cluster their sequences phylogenetically with the panzootic group of 2344b viruses and conclude on the necessity of improved biosecurity measures for poultry holdings in the Philippines.

Very few data, let alone sequences, are available from HPAI outbreaks in the Philippines. Therefore, the data are valuable, although in comparison to other publications in the same field not very extensive. Thus, a short communication format may be even more appropriate here (e.g. Table 1 can easily be moved to supplementals). In order to round up their manuscript some more data should be included here notwithstanding the paper format:

11. Have the authors retrieved all the available sequences for Philippine viruses from the public databases? It would be interesting to see how the 2023 sequences related to those of previous outbreaks in the country. At least from 2022, there are two sequences from the Philippines available in the Epiflu database, and these need to be included. Also, the tree shown can be shortened to the 2344b clade only.

22. Genotype and genome constellations play an important role in the phylo-epidemiology of gs/GD HPAI viruses. Although the authors analyse internal segments with respect to the nearest neighbors, it would be interesting to see to what genotypes (as defined in particular by Chinese papers for the southeast Asian world), have been found in the Philippines.

33. The deduction about the entry source is only marginally satisfying. No wild bird data have been published from the Philippines, so it is not known when and what HPAI viruses have been present in these populations. Also, no species of wild birds is mentioned here. Probably there is no poultry farm worldwide where not wild birds (sparrows, swallows etc.) are seen from time to time, so this cannot be taken as evidence without further stratification (and examination). On the other hand, the authors do not report on tracing-on and tracing-back investigations of the outbreak case which could elucidate further entry routes originating in poultry trade and traffic. These data must be added here to lift the manuscript to publication standard.

Comments on the Quality of English Language

English is fine.

Author Response

The authors have analysed the whole genome sequence of HPAIV H5N1 2344b from two chickens of an outbreak in layers in central Luzon, Philippines. They cluster their sequences phylogenetically with the panzootic group of 2344b viruses and conclude on the necessity of improved biosecurity measures for poultry holdings in the Philippines.

Very few data, let alone sequences, are available from HPAI outbreaks in the Philippines. Therefore, the data are valuable, although in comparison to other publications in the same field not very extensive. Thus, a short communication format may be even more appropriate here (e.g. Table 1 can easily be moved to supplementals). In order to round up their manuscript some more data should be included here notwithstanding the paper format:

Response: Thank you for the comment. We agree that the amount of sequence data we report in our manuscript is less than in other publications. Nonetheless, we surmise that since little is known about H5 HPAIVs in the Philippines, offering a manuscript that contain metadata and a simple phylogenetic analysis would be a valuable datapoint to the collective body of knowledge on H5 HPAIVs. As the reviewer has suggested, Table 1 was moved as a supplementary material and we made additional modifications to the manuscript.

  1. Have the authors retrieved all the available sequences for Philippine viruses from the public databases? It would be interesting to see how the 2023 sequences related to those of previous outbreaks in the country. At least from 2022, there are two sequences from the Philippines available in the Epiflu database, and these need to be included. Also, the tree shown can be shortened to the 2344b clade only.

Response: When the manuscript was initially submitted, the only publicly available H5 high pathogenicity avian influenza virus sequences from the Philippines in the GISAID and Genbank databases are from our group with isolate numbers EPI_ISL_19061458 (A/chicken/Philippines/BA-MHN/2022) and EPI_ISL_19057794 (A/chicken/Philippines/BA-PTY/2022). We have added the GISAID numbers to the manuscript (lines 100-102, 263-265). To my understanding (Christina Leyson), these sequences were in my unreleased files during the initial submission of this manuscript. The entries on GISAID for PTY and MHN require an edit on the “Submitting lab” entry, which is currently set as my previous institution and should be changed to Bioassets where the samples were processed and sequenced. I am currently unable to edit this on my end and will continue with GISAID to fix this. Before submitting the manuscript for review, I contacted GISAID to make the revision but did not hear a response. In order to expedite the review process, we opted to submit the sequences to GenBank and report the GenBank accession numbers in the first manuscript submitted for review.

In the GISAID and NCBI, there are many sequences of influenza virus isolates from humans originating in the Philippines. Since these were human isolates from different subtypes, we did not consider comparing these sequences to that of PTY and MHN.

Upon checking GISAID prior to submitting this revision, we now observe H5 avian influenza virus sequences from the Philippines that were submitted on July 30, 2024. These sequences were from prior outbreaks in Pampanga during 2019-2020 and are from distinct clades in the Goose/Guandong lineage. We discuss these sequences in lines 214-221. Since these were from a different outbreak and from a different lineage, we opted to continue focusing our phylogenetic analysis on clade 2.3.4.4b viruses.

Lastly, as the reviewer has suggested, we have shortened Figure 1 to only show 2.4.4.4b and also 2.3.4.4c as a frame of reference.

  1. Genotype and genome constellations play an important role in the phylo-epidemiology of gs/GD HPAI viruses. Although the authors analyse internal segments with respect to the nearest neighbors, it would be interesting to see to what genotypes (as defined in particular by Chinese papers for the southeast Asian world), have been found in the Philippines.

Response: As also suggested the reviewer, we have made additional phylogenetic analysis comparing H5 HPAIVs. We have identified PTY and MHN as belonging to the G2c group with reassortments in PB2, PB1, PA, NP, and NS. Please see Figure 1B, Supplementary Figure S1 and lines 174-180.

  1. The deduction about the entry source is only marginally satisfying. No wild bird data have been published from the Philippines, so it is not known when and what HPAI viruses have been present in these populations. Also, no species of wild birds is mentioned here. Probably there is no poultry farm worldwide where not wild birds (sparrows, swallows etc.) are seen from time to time, so this cannot be taken as evidence without further stratification (and examination). On the other hand, the authors do not report on tracing-on and tracing-back investigations of the outbreak case which could elucidate further entry routes originating in poultry trade and traffic. These data must be added here to lift the manuscript to publication standard.

Response: We greatly appreciate the reviewer’s input and comments. Based on personal communication with the farm owners, wild birds, such as egrets have been seen around the farm prior to the HPAI case. This information has been added to the manuscript on lines 138-139. Unfortunately, this is all the information that we have at the moment. Currently, there is a lack of resources to undertake epidemiological investigations around the H5N1 outbreak that we describe in the manuscript. Moreover, the case was two years ago, and it is difficult to trace back the circumstances around the case. Due to the lack of additional information, we have modified the manuscript to be more circumspect when we talk about the origins of the virus (lines 169-191).

Round 2

Reviewer 1 Report

Comments and Suggestions for Authors

Dear authors

I hope all of you are always fine. Regarding the revision of the corrected manuscript No. pathogens-3031520, titled “Molecular characterization of a clade 2.3.4.4b H5N1 high pathogenicity avian influenza virus from a 2022 outbreak in layer chickens in the Philippines” (V2). A major comment should be replied.

Major Comments

-       Phylogenetic trees for the other 7 segments (PB2, PB1, PA, NP, NA, M, and NS) should be added.

-       I recommend to separate results from discussion and to add more details about the clade 2.3.4.4b H5Nx in your country with regards to other Asian, African, European and American countries.

Author Response

Response to Reviewer 1

Dear authors
I hope all of you are always fine. Regarding the revision of the corrected manuscript No. pathogens-3031520, titled “Molecular characterization of a clade 2.3.4.4b H5N1 high pathogenicity avian influenza virus from a 2022 outbreak in layer chickens in the Philippines” (V2). A major comment should be replied.
Major Comments
- Phylogenetic trees for the other 7 segments (PB2, PB1, PA, NP, NA, M, and NS) should be added.

We have added the branches for each segment containing the PTY and MHN sequences to the main text of the article as an additional figure (Figure 2). Only a portion of the phylogenetic tree is presented for emphasis and simplicity. Nonetheless, the complete phylogenetic trees for all segments have been remade and made available in the Supplementary Figure S1. We hope that the reviewer will find this satisfactory.

- I recommend to separate results from discussion and to add more details about the clade 2.3.4.4b H5Nx in your country with regards to other Asian, African, European and American countries.

The Results and Discussion paragraphs have been separated as the reviewer recommended.

We have remade phylogenetic trees using the genotype classification by Fusaro et al [1]. These reference sequences include isolates from Asia, Africa, and Europe. The incursion of clade 2.3.4.4b viruses in the Americas have only occurred recently. The genotypes associated with this incursion are largely derived from European viruses carried through the transatlantic route [2-4].

With the addition of the reference sequences from [1], we found that the viruses circulating in Europe from 2020-2022 are genotypically distinct from our isolates PTY and MHN. Notwithstanding, we found that PTY and MHN share common ancestors with the 2020-2022 viruses from Europe at the NP, NA, and MP segments. This finding highlights the rich diversity of avian influenza viruses across the globe. A paragraph in the Results section have been added at lines 181-191.

The use of reference sequences from [1] did not alter the observation that the PTY and MHN viruses from the Philippines have close genetic relationship to that of G2c group of viruses as identified by colleagues in Japan [5, 6]. We hope the reviewer finds this additional analysis satisfactory.

References

  1. Fusaro, A.; Zecchin, B.; Giussani, E.; Palumbo, E.; Agüero-García, M.; Bachofen, C.; Bálint, Á.; Banihashem, F.; Banyard, A. C.; Beerens, N.; Bourg, M.; Briand, F. X.; Bröjer, C.; Brown, I. H.; Brugger, B.; Byrne, A. M. P.; Cana, A.; Christodoulou, V.; Dirbakova, Z.; Fagulha, T.; Fouchier, R. A. M.; Garza-Cuartero, L.; Georgiades, G.; Gjerset, B.; Grasland, B.; Groza, O.; Harder, T.; Henriques, A. M.; Hjulsager, C. K.; Ivanova, E.; Janeliunas, Z.; Krivko, L.; Lemon, K.; Liang, Y.; Lika, A.; Malik, P.; McMenamy, M. J.; Nagy, A.; Nurmoja, I.; Onita, I.; Pohlmann, A.; Revilla-Fernández, S.; Sánchez-Sánchez, A.; Savic, V.; Slavec, B.; Smietanka, K.; Snoeck, C. J.; Steensels, M.; Svansson, V.; Swieton, E.; Tammiranta, N.; Tinak, M.; Van Borm, S.; Zohari, S.; Adlhoch, C.; Baldinelli, F.; Terregino, C.; Monne, I., High pathogenic avian influenza A(H5) viruses of clade 2.3.4.4b in Europe-Why trends of virus evolution are more dilicult to predict. Virus Evol 2024, 10, (1), veae027.
  1. Youk, S.; Torchetti, M. K.; Lantz, K.; Lenoch, J. B.; Killian, M. L.; Leyson, C.; Bevins, S. N.; Dilione, K.; Ip, H. S.; Stallknecht, D. E.; Poulson, R. L.; Suarez, D. L.; Swayne, D. E.; Pantin- Jackwood, M. J., H5N1 highly pathogenic avian influenza clade 2.3.4.4b in wild and domestic birds: Introductions into the United States and reassortments, December 2021– April 2022. Virology 2023, 109860.

  2. Erdelyan, C. N. G.; Kandeil, A.; Signore, A. V.; Jones, M. E. B.; Vogel, P.; Andreev, K.; Bøe, C. A.; Gjerset, B.; Alkie, T. N.; Yason, C.; Hisanaga, T.; Sullivan, D.; Lung, O.; Bourque, L.; Ayilara, I.; Pama, L.; Jeevan, T.; Franks, J.; Jones, J. C.; Seiler, J. P.; Miller, L.; Mubareka, S.; Webby, R. J.; Berhane, Y., Multiple transatlantic incursions of highly pathogenic avian influenza clade 2.3.4.4b A(H5N5) virus into North America and spillover to mammals. Cell Reports 2024, 43, (7).

  3. Jimenez-Bluhm, P.; Siegers, J. Y.; Tan, S.; Sharp, B.; Freiden, P.; Johow, M.; Orozco, K.; Ruiz, S.; Baumberger, C.; Galdames, P.; Gonzalez, M. A.; Rojas, C.; Karlsson, E. A.; Hamilton- West, C.; Schultz-Cherry, S., Detection and phylogenetic analysis of highly pathogenic A/H5N1 avian influenza clade 2.3.4.4b virus in Chile, 2022. Emerging Microbes & Infections 2023, 12, (2), 2220569.

  4. Takadate, Y.; Mine, J.; Tsunekuni, R.; Sakuma, S.; Kumagai, A.; Nishiura, H.; Miyazawa, K.; Uchida, Y., Genetic diversity of H5N1 and H5N2 high pathogenicity avian influenza viruses isolated from poultry in Japan during the winter of 2022–2023. Virus Research 2024, 347, 199425.

  5. Mine, J.; Takadate, Y.; Kumagai, A.; Sakuma, S.; Tsunekuni, R.; Miyazawa, K.; Uchida, Y., Genetics of H5N1 and H5N8 High-Pathogenicity Avian Influenza Viruses Isolated in Japan in Winter 2021–2022. Viruses 2024, 16, (3), 358.

Reviewer 2 Report

Comments and Suggestions for Authors

I have no additional questions about the updated version of the paper. The replies to the reviewer's comments are satisfactory for further consideration of the paper.

Author Response

Thank you for your comments!

Reviewer 3 Report

Comments and Suggestions for Authors

The authors have answered the editorial questions and comments adequately and processed them satisfactorily.

Author Response

Thank you for your comments!